# Transfection Models to Investigate *Plasmodium vivax*-Type Dormant Liver Stage Parasites

**DOI:** 10.3390/pathogens12091070

**Published:** 2023-08-22

**Authors:** Annemarie Voorberg-van der Wel, Anne-Marie Zeeman, Clemens H. M. Kocken

**Affiliations:** Department of Parasitology, Biomedical Primate Research Centre, 2288 GJ Rijswijk, The Netherlands; zeeman@bprc.nl (A.-M.Z.); kocken@bprc.nl (C.H.M.K.)

**Keywords:** malaria, hypnozoite, *Plasmodium vivax*, *Plasmodium cynomolgi*, transfection, genetic modification, liver stage

## Abstract

*Plasmodium vivax* causes the second highest number of malaria morbidity and mortality cases in humans. Several biological traits of this parasite species, including the formation of dormant stages (hypnozoites) that persist inside the liver for prolonged periods of time, present an obstacle for intervention measures and create a barrier for the elimination of malaria. Research into the biology of hypnozoites requires efficient systems for parasite transmission, liver stage cultivation and genetic modification. However, *P. vivax* research is hampered by the lack of an in vitro blood stage culture system, rendering it reliant on in vivo-derived, mainly patient, material for transmission and liver stage culture. This has also resulted in limited capability for genetic modification, creating a bottleneck in investigations into the mechanisms underlying the persistence of the parasite inside the liver. This bottleneck can be overcome through optimal use of the closely related and experimentally more amenable nonhuman primate (NHP) parasite, *Plasmodium cynomolgi,* as a model system. In this review, we discuss the genetic modification tools and liver stage cultivation platforms available for studying *P. vivax* persistent stages and highlight how their combined use may advance our understanding of hypnozoite biology.

## 1. Introduction

Malaria continues to affect a major proportion of the global human population. Data from the World Health Organization (WHO) indicate that the number of malaria cases show a rising trend, with 232 million cases in 2019, 245 million cases in 2020 and an estimated 247 million malaria cases in 2021 in 84 malaria-endemic countries [1]. More than 600,000 people died from the disease, with most deaths (76%) occurring in children aged under five years [1]. Malaria is caused by protozoan parasites of the genus Plasmodium. Of the five species known to infect humans, by far the deadliest is *Plasmodium falciparum.* The second most important causative agent of human malaria is *Plasmodium vivax.* Even though morbidity and mortality numbers are lower than those attributed to *P. falciparum*, almost half of the world’s population is at risk of getting infected with *P. vivax.* Most cases have been recorded in Southeast Asia, the Western Pacific regions and the Americas [1,2]. Recent reports also indicate widespread transmission of *P. vivax* in previously presumed resistant Duffy-negative people in Sub-Saharan Africa [3,4,5]. Infection with *P. vivax* is not trivial, as it can be severe, and it may even lead to death [6,7]. Especially young children and pregnant women are at risk in endemic areas [8].

*P. vivax* possesses some biological traits that promote a geographically widespread transmission and persistence inside the host. The most striking features are (i) early sexual stage development enabling transmission prior to clinical symptoms [9] and (ii) the development of dormant stages that can remain without symptoms inside the liver for weeks to years before resuming growth, resulting in a fresh blood stage infection (relapse) and transmission [10]. *P. vivax* preferentially invades immature reticulocytes that are generally restricted to the bone marrow, resulting in low percentages of blood stage infection [11,12]. Hence, it has been suggested that relapse infections originally may have evolved as a mechanism to enable competition with more virulent *Plasmodium* species [13]. Later in evolution, further adaptation for prolonged persistence in the liver before fresh blood stage infection (‘long latency phenotype’) may have enabled the parasite to survive in temperate areas with shorter mosquito transmission seasons [14]. In addition to relapse infections resulting from hypnozoite activation, it has been suggested that *P. vivax* parasites accumulating in the spleen and bone marrow may serve as an important reservoir that might be the source of recrudescence, contributing to chronic malaria [15,16,17,18].

Hypnozoites are difficult to target with currently available drugs. Only the 8-aminoquinolines primaquine and tafenoquine have been approved as radical cure treatment drugs. However, these drugs can cause severe hemolysis in patients with glucose-6-phosphate dehydrogenase (G6PD) deficiency [19,20,21]. G6PD deficiency occurs in about 8% of people living in malaria-endemic countries [22]. These problems, and a limited access to G6PD tests in resource-constrained settings, have precluded the widespread use of the drugs, especially in pregnant or lactating women and young infants [23].

Hypnozoites were discovered decades after the other stages of the Plasmodium life cycle [24], and hypnozoite research is challenging for multiple reasons. Firstly, the location of hypnozoites inside the liver is largely inaccessible for experimentation. Secondly, only a subset of primate malaria parasite species form hypnozoites [25], restricting in vivo studies to NHPs. Last but not least, *P. vivax* has received little to no attention whereas the more dangerous *P. falciparum* has received the vast majority of funds for study [26]. With the elimination of malaria on the agenda [27], it has been recognized that increased research efforts and new areas of investigation (e.g., diagnostic tools for tissue-residing quiescent parasites) are needed to tackle the complexities of the life cycle of *P. vivax*-type parasites [26]. During the past decade, various studies either using the human malaria parasite *P. vivax* or the phylogenetically closely related NHP parasite *Plasmodium cynomolgi*, have unveiled a first glimpse of hypnozoite biology (reviewed below). This was enabled by technical advances in hypnozoite culture systems, the availability of genome sequences of *P. vivax-*type parasites and transcriptomics studies that identified genes that may be involved in hypnozoite biology. Follow-up studies dissecting the functional roles of the identified genes will greatly benefit from sophisticated genetic engineering systems.

The development of technology to genetically manipulate Plasmodium parasites [28,29] has been a key tool in a wide range of research areas. It has been instrumental in gaining an understanding in biological processes underlying the biology of different stages of the malaria parasite. For example, this technology has identified factors necessary for the regulation of gametocyte development [30,31] and enabled investigations into the functional roles of proteins involved in red blood cell invasion [32]. Furthermore, it has been implemented as tool in drug and vaccine development. For example, it has been used to create genetically attenuated parasites for experimental vaccines [33,34] as a new promising vaccine strategy. Furthermore, the development of bioluminescent Plasmodium reporter lines has enabled miniaturization of drug screening platforms, greatly improving efficiency of screening drugs against different stages of the parasite [35].

Genetic engineering platforms have been developed for a range of Plasmodium species [28,29,35,36,37,38,39]. Stable transfection of malaria parasites traditionally relies on a source of blood stage parasite material for the introduction of DNA, a procedure for introducing the DNA and a method to select for parasites that have taken up the DNA. Depending on the parasite species, this can be accomplished in vitro (*P. falciparum*) [29], in vivo (*P. vivax*) [40,41] and the rodent malarias *P. berghei*, *P. yoelii*, *P. chabaudi* [28,36,38] or both (the NHP parasites *P. knowlesi* [42] and *P. cynomolgi* [37,43]) (Figure 1). Besides differences in in vitro/in vivo dependence, the level of advancement of developed tools differs between species. At the same time, species-specific biological differences may complicate the extrapolation of findings to other species. For example, between species clear differences exist in gametocyte biology, not only at the level of morphology, maturation rate and longevity, but also in the genes involved in gametocyte development [44].

Investigations into the biology of hypnozoites are particularly hampered by the paucity of model systems. *P. vivax* genetic engineering has been limited to blood stage parasites. To date, the only model system that has enabled hypnozoite research with genetically engineered parasites is the NHP *P. cynomolgi* [45,46,47,48,49,50]. In this review, we describe the strengths, weaknesses and opportunities of the different model systems available for *P. vivax* research in the context of the application of genetic manipulation technology to hypnozoite research.

## 2. Liver Stage Models for *P. vivax*

### 2.1. P. vivax In Vivo Liver Stage Models

The presence of *P. vivax* hypnozoites was originally demonstrated in liver biopsies of chimpanzees that had been experimentally infected with *P. vivax* sporozoites [10]. *P. vivax* only develops in humans and apes, and has been adapted to grow in some New World monkeys [51,52]. Combined with the inaccessible location of hypnozoites and their low quantities inside the liver, this complicates liver stage research. Hence, studies investigating liver stage biology in situ have been limited until the seminal work of Mikholajczak et al. [53] demonstrated that complete *P. vivax* liver stage development could be achieved in a severely immunocompromised FRG KO mouse model transplanted with human hepatocytes. A detailed phenotypic analysis of the liver stage parasites after different time points of infection demonstrated the presence of two distinct parasite populations inside the liver. The small stages were posited to be hypnozoites given their single nucleus, persistence throughout the 21-day time course of the study and differential insensitivity to the schizonticide atovaquone in comparison to growing liver stages. The hypnozoites were shown to slowly increase in size over time, indicating metabolic activity [53]. Following this work, this mouse model has been applied to study the impact of a partially protective *P. vivax* pre-erythrocytic vaccine on reducing relapses. It was shown that although the vaccine regimen did not completely prevent primary infection, treated mice experienced 62% fewer relapses [54]. Furthermore, another study demonstrated that by first eliminating liver stage schizonts through a schizont-specific drug treatment, recurrence of secondary schizonts could be assessed by microscopy and qRT-PCR, enabling assaying the activity of anti-relapse drugs [55]. Interestingly, the hypnozoite formation frequency in the humanized mouse model was reported to be lower than that observed for *P. vivax* in vitro platforms [55]. This could be a result of sub-optimal artificial in vitro culture conditions, or differences in parasite clearance rates between in vitro versus in vivo parasites. Since the different platforms use *P. vivax* sporozoites derived from field isolates, variation in parasite characteristics may also influence the hypnozoite ratios. A side-by-side comparison of the different *P. vivax* in vitro and in vivo models could possibly give more clues to this.

### 2.2. P. vivax In Vitro Liver Stage Platforms

*P. vivax* selectively invades younger reticulocytes, and as a result, the development of a robust long-term in vitro blood stage culture system has proven cumbersome [56,57]. Hence, mosquito transmission needed for liver stage investigations can only be achieved with in vivo-derived material from isolates from infected human subjects from endemic areas, or experimentally infected NHPs (with specifically adapted parasite strains) as a source for sexual stages. Despite these complications, a wide variety of in vitro liver stage platforms has been developed to study hypnozoite biology and to assay drug activity against these stages—reviewed in [58]. This includes the infection of various types of hepatoma cell lines [59,60], immortalized human hepatocytes (HC-04) [61] or hepatocyte-like cell lines (imHC) [62] and iPSC-derived hepatocyte-like cells (iHCLs) [63]. In addition, primary human hepatocytes (PHH) have been used in different formats: in a coculture system [64,65], as 3D spheroids [66], or in 2D monolayer formats [67,68].

Using micropatterned primary human hepatocytes co-cultured with murine embryonic fibroblasts (MPCC) infected with *P. vivax* field isolates from Thailand, complete liver stage development was recapitulated in vitro including subsequent reticulocyte infection [64]. Both hypnozoites and liver stage schizonts were observed and characterized at the molecular level. This type of work is challenging, given the presence of liver stage parasites at a low infection rate (which differs between platforms*)* and a 100-fold smaller genome [69], hence lower amounts of transcripts, compared to the genome of the host cell [70]. These issues were counteracted by enrichment of parasite transcripts using custom-made baits against the *P. vivax* P01 genome. Furthermore, bulk transcriptomics of hypnozoite-enriched samples [64] was achieved by treatment with a PI4K inhibitor known to selectively kill liver stage schizonts [71]. This showed that samples enriched for hypnozoites exhibited reduced transcriptional activity and had lower levels of transcripts encoding functions related to cell division and invasion machinery compared to untreated samples containing both hypnozoites and schizonts, indicative of a quiescence state of hypnozoites [64]. A follow-up of this work using the same platform in combination with Seq-Well single-cell sequencing enabled a more refined transcriptomic analysis of both parasite and host cells [72]. This indicated that hypnozoites can persist in a quiescent state through transcriptional or translational repression and rely on proteolytic activity to remain viable. Furthermore, the results revealed that a subset of sporozoites developing in hepatocytes with reduced metabolism may already be committed for sexual development [72]. Another study investigating *P. vivax* parasite and host cell transcriptomes at single-cell resolution used primary hepatocytes infected with Cambodian *P. vivax* parasites combined with a high-throughput droplet-based single cell RNA-seq workflow [73]. The results from this liver stage platform with high infection rates (5.50–6.93% *P. vivax*-infected hepatocytes at day five post sporozoite infection), underscored the notion that hypnozoites are metabolically active, albeit at low levels, and found evidence for the existence of hypnozoites in different transcriptomic states. The authors hypothesized that this may reflect heterogeneity in phenotypes ranging from persisting to activating hypnozoites. In the ‘persister’ subgroup of hypnozoites, differential expression of genes involved in post-transcriptional/translational repression was observed, suggesting that these mechanisms may underly quiescence.

### 2.3. P. cynomolgi Liver Stage Research Platforms

*P. cynomolgi* is an NHP malaria parasite that is phylogenetically closely related to *P. vivax* and shares important characteristics of *P. vivax* including hypnozoite formation [74,75] and early gametocyte development [76]. Notably, the existence of hypnozoites as developmental stage of the life cycle in relapsing parasite species was for the first time demonstrated in biopsies of macaques infected with *P. cynomolgi* sporozoites [77]. The demonstration that drug activity profiles were highly similar between *P. cynomolgi* and *P. vivax* led in the 1980s to the incorporation of large-scale drug screening studies with *P. cynomolgi* sporozoite-induced infections in rhesus monkeys as a central step in efforts (which also used patients undergoing *P. vivax* malaria therapy as well as prison inmate volunteers [78]) to find new drugs effective against hypnozoites [79,80,81,82,83,84,85,86]. In addition to the opportunity to study relapse infections in vivo, an in vitro liver stage culture system was developed for *P. cynomolgi* [87]. Recently, this in vitro system has been revived and adapted for drug screening of anti-hypnozoite compounds, setting the stage for the development of similar drug screening systems for *P. vivax* [88,89]. Furthermore, the availability of these in vitro systems has facilitated access to hypnozoites for biological studies.

Routinely, liver stage culture platforms of *P. cynomolgi* use monolayers of primary macaque hepatocytes in 96- or 384-well format [50,90]. In these cultures, generally ±60% of *P. cynomolgi* M liver stage parasites are present as hypnozoites [89,91]. Cultures infected with *P. cynomolgi* B strain parasites appeared to contain a higher number of hypnozoites, which may reflect strain dependent variation (although it is not certain whether the M and B lines used were truly separate lines—[92]), or differences in culture conditions and/or hepatocyte cells. Full liver stage development of *P. cynomolgi* M liver stages was demonstrated in vitro including RBC invasion [49]. By applying a Matrigel cover to 2D monolayer cultures, *P. cynomolgi* liver stage cultures can be maintained for prolonged periods of time, providing the opportunity to study hypnozoite reactivation in vitro [93]. Hypnozoite reactivation was inferred from the observation that liver stage parasites emerged from cultures that had been drug treated to remove liver stage schizonts [93]. Using microdissection of *P. cynomolgi-*infected liver cells, a first glimpse of the transcriptome of hypnozoites and liver schizonts was reported [94]. The authors suggested a potential role for an AP2-transcription factor which was termed AP2-Q in hypnozoite formation. The function of this transcription factor awaits functional validation.

In addition to 2D culture capabilities, a 3D spheroid-cultured hepatocyte system has been developed to better preserve hepatocyte functionality in vitro [66]. Imaging of Plasmodium-infected hepatocytes proved difficult in this system and only possible by time-consuming and data-intensive imaging stacking methods. However, following dissociation, *P. cynomolgi* hypnozoites and liver schizonts obtained from the spheroid cultures could be detected. Furthermore, the system enabled recapitulation of the full life cycle (liver stages, transmissible stages, and asexual blood stages) as evidenced by the appearance of infected RBC after overlaying the spheroid-cultured hepatocytes with blood [66].

## 3. Transfection Technology to Study Hypnozoite Biology

### 3.1. P. vivax Genetic Engineering Technology

Functional analysis of mechanisms involved in hypnozoite biology depends on the availability of sophisticated tools for genetic engineering. For *P. vivax,* this type of tools is limited. Only two studies have reported proof-of-concept of the ability to genetically manipulate *P. vivax* [40,41]. In the absence of a *P. vivax* in vitro blood stage culture system, this type of work can only be performed in vivo. Hence, both transfection studies used splenectomized *Saimiri boliviensis* monkeys as donor to provide blood stage *P. vivax* as a source for transfection.

In the first proof of successful *P. vivax* transfection, *P. vivax* Sal I blood stage trophozoites were obtained from a donor monkey [41]. Following Percoll purification of trophozoite-infected cells, parasites were electroporated with bioluminescence reporter constructs containing the *firefly* or *renilla* luciferase gene controlled by heterologous 5′ UTRs (*hrp3* or *cam*) and 3′ UTRs (*hrp2* or *cam*) of *P. falciparum* genes. Following transfection, parasites were maintained in 22 h short-term cultures before luciferase measurements. This transient transfection system was successful as evidenced by detection of luciferase signals specifically in extracts from transfected parasites, indicating that these heterologous *P. falciparum* UTRs display activity in *P. vivax.* Transient transfection can only be used in short-term studies investigating gene expression, since there is no selection for parasites harboring the constructs and thus plasmids are lost over time as the parasites proliferate.

Nine years after this first publication, stable transfection of *P. vivax* was reported in which long-term maintenance of the constructs was achieved using a selectable marker and recipient monkeys to enable selection of drug resistant parasites [40]. Splenectomized *Saimiri boliviensis* donor monkeys were infected with the pyrimethamine-sensitive *P. vivax* Chesson line and parasites were electroporated with constructs carrying a pyrimethamine resistant *P. vivax dihydrofolate reductase (Pvdhfr)* gene and zinc-finger nucleases (ZFNs) to target constructs to the wild type *P. vivax dihydrofolate reductase (Pvdhfr)* locus. In contrast to the transient transfection study, parasites were not purified, but for each electroporation a leukocyte depleted RBC pellet obtained from a 3 mL blood sample containing *P. vivax* blood stages at a ±0.5% parasitemia was used. The developmental stage of the parasites at the time of harvesting was not mentioned. Following ex vivo electroporation, parasites were inoculated into four splenectomized *Saimiri boliviensis* recipient monkeys. In one of four recipient monkeys, successful editing of the *Pvdhfr* was shown. An explanation for this suboptimal success rate was that parasitemias in the primates over several weeks are highly variable, depending on susceptibility to infection and immune responses, and can clear spontaneously without treatment [40].

### 3.2. P. vivax Hypnozoite Research Benefits and Limitations

Obviously, using *P. vivax* itself as model system is the ideal situation. Robust in vitro [64,68] and in vivo [55] liver stage models are available for studying hypnozoite biology. A major hurdle for *P. vivax* is the limited arsenal of genetic engineering tools to study functional aspects of hypnozoite biology. This is largely due to the lack of an in vitro blood stage culture system; hence genetic engineering of *P. vivax* has been fully reliant on the use of in vivo blood stage parasites from infected NHP donors and recipients. Due to the insusceptibility of macaques to *P. vivax* parasites [95], the two proof-of-concept studies reporting transfection of *P. vivax* depended on experimentally infected available New World monkeys [40,41]. Hence, ethical reasons, costs and limited availability of these NHPs complicate this type of experimental studies. One way to overcome this may be to attempt using humanized mice as donor and recipients for genetic engineering of *P. vivax* or to achieve successful in vitro blood stage cultivation of *P. vivax.*

### 3.3. P. cynomolgi Genetic Engineering Technology

#### 3.3.1. *P. cynomolgi* In Vivo Genetic Engineering Technology

Successful transfection of *P. cynomolgi* was demonstrated in 1999 in a proof-of-concept study in which the pyrimethamine sensitive *P. cynomolgi* M line was transfected with episomal constructs that included a pyrimethamine resistant *dhfr* gene to enable pyrimethamine selection of parasites that have taken up DNA in vivo in recipient monkeys [37]. Blood stage schizont-infected RBC obtained from a donor monkey were ex vivo electroporated with plasmids with a selectable marker gene controlled by UTRs of *P. berghei* or *P. falciparum* (the genome of *P. cynomolgi* was published years later [96,97]). The material was then pooled and injected into a recipient monkey. After pyrimethamine selection in vivo, a resistant parasite population emerged. Only plasmids containing the *P. berghei* UTRs were detected in resistant parasites, indicating that *P. berghei dhfr-ts* flanking regions may more efficiently control selectable marker gene expression than *P. falciparum hrp2*/*hrp3* regions. A second transfection experiment used this technology to functionally investigate a 95 kDa protein located in the caveola–vesicle complexes, the CVC protein (*pcyphist*/*cvc-81_95_)*, present in both *P. vivax* and *P. cynomolgi* [98]. Attempts to target this gene by double crossover recombination using a linearized construct with a selection cassette flanked by homologous regions targeting *pcyphist*/*cvc-81_95_* failed, indicating that the gene is essential. However, Akinyi et al. did report the first example of integration of a transgene (the RFP reporter gene) into the *P. cynomolgi* genome, which was shown by the continued expression of RFP in blood stage *P. cynomolgi* after the original in vivo selection and multiple blood stage passages in rhesus monkeys [98]. PCR and plasmid rescue analyses demonstrated the presence of the reporter gene, but the absence of the plasmid backbone. Given the uneven segregation of episomes, they are rapidly lost in the absence of selection pressure [99]. This complicates liver stage research as it is challenging to maintain drug selection pressure needed to maintain episomal constructs during mosquito transmission. For expression of transgenes during liver stage development, one option is to target constructs for integration into the genome, but this is a less efficient process than episomal transfection. Furthermore, it requires a neutral locus that can be targeted without having detrimental effects in different parts of the life cycle. While such loci have been described for other malaria parasite species [100,101], a neutral *P. cynomolgi* locus is currently not known. An alternative method to enable transgene expression in liver stage parasites is to include a centromere in the transfection construct to enable a more even pattern of distribution of episomes during parasite multiplication [102]. This method was successfully applied for the first time in *Plasmodium* by Iwanaga et al. who showed that constructs containing a *Plasmodium* centromere could be efficiently transfected and maintained throughout the complete life cycle in *P. berghei* and *P. falciparum* [103]. By analogy, transfection of *P. cynomolgi* with a plasmid construct that includes a newly identified *P. cynomolgi* centromere resulted in maintenance of the construct throughout the parasite life cycle, with a mean of 66% of liver stage parasites expressing the transgenes [48] (Table 1).

#### 3.3.2. Application of *P. cynomolgi* Vivo-Derived Transgenic Parasites to the Study of Hypnozoite Biology

The centromere-containing plasmid as described in the previous paragraph was used for live visualization of liver stage parasites including hypnozoites. Using the GFP-marker expressed from this plasmid, hypnozoite- and schizont-infected hepatocytes could be separated from uninfected hepatocytes by fluorescence-activated cell sorting (FACS) [48] and subjected to transcriptomics. This enabled an analysis of host cells of infected versus uninfected cells [104] and a bulk transcriptome analysis of *P. cynomolgi* liver stages isolated at different timepoints after sporozoite infection [45,47]. It was shown that while hypnozoites progressively decrease transcription, processes involved in quiescence, energy metabolism and maintenance of genome integrity are maintained [45]. These datasets also indicated a massive upregulation of Lisp2 in liver stage schizonts compared to hypnozoites and a follow up study identified this protein as an early marker of liver stage growth [91]. Based on this differential expression pattern of Lisp2, *P. cynomolgi* dual marker parasites were developed [49,50].

The dual fluorescent *P. cynomolgi* line that included GFP driven by the constitutive promoter *hsp70* and *lisp2*-driven mCherry provided the unique opportunity to observe in real time with a fluorescence microscope the development of liver stage parasites including hypnozoites and to follow hypnozoite activation directly and unequivocally in the absence of drug treatment (that would otherwise have been needed to distinguish aborted liver stage schizonts from activated hypnozoites). Through prolonged monitoring of the dual fluorescent *P. cynomolgi* (up until day 22 post infection of the hepatocyte), almost 40 years after their discovery, it was for the first time formally demonstrated that that hypnozoites can activate and resume development after the primary liver stage schizonts have completed development and ruptured [49].

Similarly, a dual bioluminescent parasite line was developed in which the bright NanoLuc luciferase [105] was constitutively expressed using the *hsp70* promoter and Firefly luciferase was driven by the liver schizont-specific *lisp2* promoter [91]. This enabled a robust, ultrafast and sensitive detection of hypnozoites through enzymatic detection of bioluminescence [50]. This may enable the development of bioluminescence-based high-throughput systems for screening compounds with anti-hypnozoite activity.

To increase our knowledge of mechanisms underlying hypnozoite dormancy, the dual fluorescent *P. cynomolgi* parasite line described above was used for GFP-fluorescence based single cell FACS isolation of liver stage parasites at early (day 2–6) stages of parasite development [46]. Subsequent single cell RNA-sequencing analysis of the sorted parasites revealed various clusters of parasite populations with different expression patterns. In the putative ‘hypnozoite’ clusters, several genes were identified to be specifically transcribed in hypnozoites. This was validated by RNA-FISH. Strikingly, ChIP-seq of sporozoites and blood stages, and ATAC-seq of hypnozoites versus liver schizonts indicated differences in chromatin environment between these parasite forms. Given that the identified hypnozoite-specific transcripts encode putative RNA-binding proteins, it was hypothesized that these epigenetically regulated RNA-binding genes may underly hypnozoite formation and activation processes [46]. This hypothesis needs to be validated by functional studies, which require sophisticated (conditional) knock-out systems as have been developed for other Plasmodium species [106,107,108], but not yet for *P. cynomolgi* vivo parasites.

#### 3.3.3. *P. cynomolgi* Vitro Genetic Engineering Technology

For *P. cynomolgi*, an in vitro blood stage culture system has been developed [109,110]. In this system, the *P. cynomolgi* Berok line was adapted to in vitro culture in macaque red blood cells. Recently, this in vitro line was successfully used to edit the putative drug resistance marker MDR1 Y976F using CRISPR-Cas9 [43]. This shows the important proof-of-concept that transfection of the *P. cynomolgi* vitro line is feasible. However, the authors noted that the methodology still requires optimization, as the editing of *Pcymdr1* Y976F was the only successful editing event from a range of other orthologous markers of drug resistance after two years of experimentation.

It is not known if this system can be directly used for *P. cynomolgi* liver stage research. Successful transmission of the in vitro *P. cynomolgi* Berok line has been shown following an in vivo passage through a monkey [110], but transmission without a monkey passage has not yet been shown (Figure 2). However, the fact that monkey-passaged in vitro *P. cynomolgi* Berok does give rise to infectious sporozoites demonstrates that the capacity of the vitro line to develop into sexual stages has not been lost.

### 3.4. P. cynomolgi Hypnozoite Research Benefits and Limitations

Findings obtained with *P. cynomolgi* as a model for *P. vivax* will always require validation in *P. vivax* itself. This is especially important in the light of the different relapse characteristics known to exist for *P. vivax* [14]. *P. cynomolgi* B/M has short latency characteristics and can thus serve as a model for fast relapsing *P. vivax* isolates. It is not known whether findings for this frequently relapsing *P. cynomolgi* can serve as a valid model for the *P. vivax* long latency lines, and it is unknown whether long-latency *P. cynomolgi* lines exist. Another limitation of the *P. cynomolgi* system is that it depends on NHPs for transmission and liver stage research. Genetic engineering of the *P. cynomolgi vitro* parasite may overcome this [43,110], but transmission without monkey passage has not been reported yet and the blood stage vitro culture still requires macaque red blood cells. Besides these limitations, the *P. cynomolgi* models provide a unique opportunity: a combined use of robust transfection methods and liver stage cultures enables transgene expression in liver stage cultures, including hypnozoites [45,46,47,48,49,50] (Figure 2). Further development of this technology to include more sophisticated genetic engineering tools (e.g., DiCre [106] and Crispr/Cas9 [111] technologies) as have been described for other Plasmodium species is warranted to enable more detailed investigations into mechanisms underlying hypnozoite dormancy and activation.

## 4. Conclusions and Outlook

Functional analysis of genes involved in hypnozoite biology is complicated. Current model systems each have their benefits and limitations. Robust liver stage culture platforms enabling the study of *P. vivax* hypnozoites are in place, but in the absence of a blood stage culture system rely on patient or NHP-derived parasites. Transfection of *P. vivax* is cumbersome and has not been applied to liver stage research. *P. cynomolgi* is an excellent model for *P. vivax.* As with any model system, biological findings in the model need validation in *P. vivax,* and it depends on NHPs, but robust liver stage culture systems including hypnozoites are available. Combined with the capacity to genetically manipulate this species, *P. cynomolgi* is currently the only parasite for which genetic engineering has been applied to study hypnozoite biology.

Through synergistically exploiting the strengths of each model, the complexities associated with *P. vivax*-type hypnozoite research may be overcome. A variety of tools for the genetic engineering of another closely related NHP parasite, *P. knowlesi*, is available [112,113,114,115,116,117,118,119,120]. However, this parasite species does not appear to develop into hypnozoites, since no hypnozoites were detected in macaques that had been infected with *P. knowlesi* sporozoites, in contrast to *P. cynomolgi-*infected rhesus monkeys [121]. Nonetheless, it can be envisaged, for example, that *P. knowlesi* can be used to optimize transfection conditions and to validate genes potentially involved in hypnozoite development (i.e., overexpression of a candidate gene may induce hypnozoite-like features in *P. knowlesi* liver stage parasites). *P. cynomolgi* genetic engineering enables direct manipulation and investigation in candidate genes involved in hypnozoite biology. Validation of findings may eventually be performed in *P. vivax.* Making optimal use of available and improved model systems may finally provide new insights in the cryptic hypnozoite biology.

## Figures and Tables

**Figure 1 pathogens-12-01070-f001:**
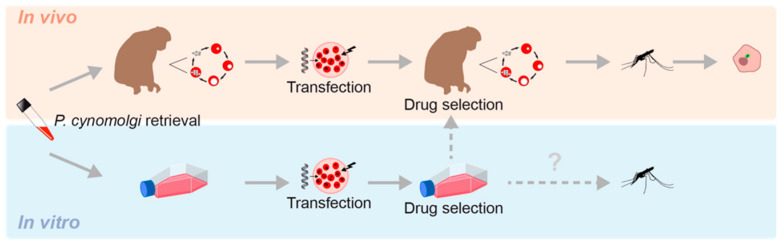
*P. cynomolgi* in vivo and in vitro transfection capabilities. Transgenic parasites can be drug selected in vivo or in vitro. To date, successful transmission of *P. cynomolgi* has only been described following a monkey passage (broken arrow) and it is not known whether in vitro cultured parasites can be transmitted in the absence of a monkey passage (broken arrow with question mark). Transgenic liver stage parasites, including hypnozoites (upper right-hand side, schematic drawing of a hepatocyte with GFP expressing hypnozoite) can be studied in vitro following sporozoite inoculation of cultured primary hepatocytes.

**Figure 2 pathogens-12-01070-f002:**
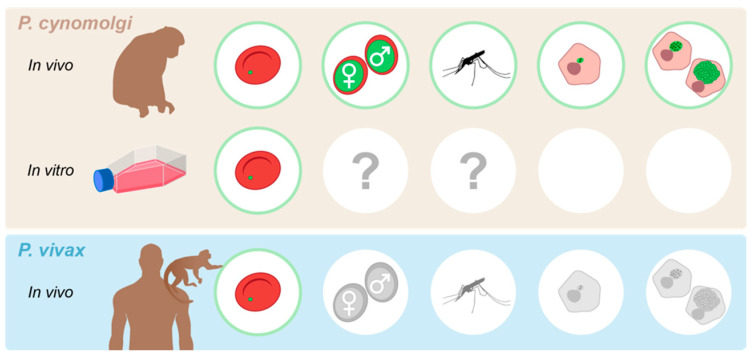
Access to different stages of the life cycle of *P. vivax*-type parasites. Stages of the parasite life cycle for which the availability of transgenic parasites has been described are depicted in green; stages that are accessible for experimentation, but that have only been studied using wild-type parasites are depicted in grey. Question marks highlight the current lack of knowledge on the capacity for in vitro cultured parasites to be transmitted without passage through a monkey.

**Table 1 pathogens-12-01070-t001:** Genetic engineering tools developed for the study of *P. vivax*-type parasites.

	Material Used for Transfection	Transfection Tools Available	Used to Investigate	Refs
*P. vivax* vivo	In vivo-derived blood stages	Transient transfection, zinc-finger mediated recombination	Blood stages in vivo	[40,41]
*P. cynomolgi* vivo	In vivo-derived blood stages	Homology directed recombination, episomal, centromere	Blood stages in vivo; liver stage schizonts and hypnozoites in vitro	[37,45,46,47,48,49,50,98]
*P. cynomolgi* vitro	In vitro-derived blood stages	Episomal, Crispr/Cas9	Blood stages in vitro	[43]

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
