# Peer review of "Transfection Models to Investigate Plasmodium vivax-Type Dormant Liver Stage Parasites"

_pathogens, 2023, doi:10.3390/pathogens12091070_

Round 1

Reviewer 1 Report

RE:  Transfection models to investigate Plasmodium vivax-type 2 dormant liver stage parasites Manuscript comments

By reviewing such interesting approaches, the authors of the present article attempted to close the gap in the problems for malaria control with regard to the biological complexity of P. vivax. I have one or two helpful suggestions for the paper.

-Background Information needed (You must first demonstrate that malaria is still a significant threat to public health, as well as the species of parasites that cause human malaria and the parasite with these hyponozoite characteristics).

Epidemiology of malaria.

You need to prove that the number of cases of malaria grew in 2021. There were a presented 247 million cases in 2021, up from 245 million in 2020 and 232 million in 2019. This will help to show that malaria is still a big problem. Deaths are way up; cases are also up.

“More than 600,000 people died from the disease, with children under the age of five accounting for the majority (76%) of the fatalities”. To cite this, kindly do so.

-Please specify the region from which this fatality occurred as "most were young children"?  Is it from sub-Saharan Africa??

Low percentages of blood stage infections are caused by P. vivax's selective invasion of reticulocytes. Yes, the parasite preferentially infects the youngest CD71+ reticulocytes, which P. vivax preferentially invades using the CD71/transferrin receptor molecule as an invasion ligand. You should consider the most recent article. For more information, please refer to Plasmodium vivax: the possible barriers it presents to the eradication of malaria by Habtamu, Petros, and Yan. For your references, see Trop Dis Travel Med Vaccines. 2022 Dec 15;8(1):27. doi: 10.1186/s40794-022-00185-3. PMID: 36522671; PMCID: PMC9753897.

Hypnozoites are challenging for currently available medications to target. Only the 8-aminoquinolines, primaquine and tafenoquine have been licensed as radical curative treatment medicines, but their widespread usage is prohibited by their severe adverse effects, especially in children and pregnant women. Other than pregnant women, breastfeeding mothers, and infants, the patient must also have a G6PD test. However, access to the test presents additional challenges; if you have any suggestions, contribute it below.

“Finally, by far most of the malaria research funding has been directed to the more deadly P. falciparum and P. vivax has been largely neglected” Please reword the sentence as follows. “The funding for malaria research has mostly been focused on the more dangerous P. falciparum strain, while P. vivax has received little attention”. Or “last but not least, P. vivax has received little to no attention whereas the more dangerous P. falciparum has received the vast majority of funds for study” ………. not neglected.

……...it has been recognized that increased research efforts are needed to tackle the complexities of the life cycle of P. vivax-type parasites……………… this increased research efforts are not only required to address the complexity of the P. vivax-type parasites' life cycle, it rather need new thinking ( eg.  tissue silent parasite stage diagnostic tool development & “OMIC studies”) .  

P. vivax selectively invades reticulocytes……….. you better say that P. vivax invades young reticuloctes rather than young reticulocytes if you want to talk about how P. vivax selectively invades reticulocytes.

Finally, transient transfection is widely employed in short-term studies to look at the effects of up- or down-regulating a particular gene. As a result, you must mention this in your paper. This is because transgenic expression disappears over time as host cells proliferate. Unless stated as one of the shortcomings.

Reviewer 2 Report

The authors describe here the latest developments in the much-neglected field of genetic manipulation efforts in Plasmodium vivax and related parasites. I have the following minor issues:

 1.  Line 27: The first words of the paper, the disease malaria, may be modified to Malaria continues...

2.  Line 36: the reference used is a modelling study whereas the authors are suggested to cite World Malaria Report data that is a standard comparative reference. Further, it appears that Americas bear the major brunt of P. vivax which may not be the case. In addition, the data is quite old (2017).

3.  Lines 45-50: the word 'renewed' may be misleading as hypnozoites may start a fresh infection with genetically the same or a different parasite. Fresh infection may be used.

4.  Lines 53-55: the authors are suggested to mention the role of spleen in harboring the parasite reservoir (see Kho et al, NEJM) and recent papers from Markus (Trends in Parasitology 2022 and TMID 2023) for unbiased mention of hypnozoites. Further, a mention of G6PD deficiency is much needed here.

5. Line 57: Please correct 'malaria life cycle' to Plasmodium life cycle

6. Line 58: the phrase "their research" needs to be modified to hypnozoite research for clarity

7. Line 60: the use of "malaria species" is erratic as malaria is not an organism

8.  Line 66: use of the phrase "sister NHP parasite" may be replaced by a more scientific term

9. Figures are of very poor resolution

10. Line 123: the word liver is misspelled as lier

11.  Line 140: Pv selectively invades younger reticulocytes

12. Line 217: The term "malaria infected" is erratic. It should be Plasmodium-infected

13. Line 221: Please elaborate on what is meant by "full life cycle" here

14. Line 328: Please rephrase "live monitor the development"

15. Line 409: The statement that Pk does not form hypnozoites is a very strong statement and may be revised. 

The use of the English language is acceptable with minor spell checks and rephrasing.

Reviewer 3 Report

A concise but nicely summarized Review paper by top experts in this research field.

I found only one typo on line 196.

Please delete one of the duplicated "drug screening."

This review summarized the latest progress of the NHP malaria model (both in vivo and in vitro). The most updated summary is available on this research topic. The texts in Figures 1 & 2 are too small.  Please enlarge them. The Figure legends should be more detailed to self-explain to the readers.
